# A qualitative enquiry of health care workers' narratives on knowledge and sources of information on principles of Respectful Maternity Care (RMC)

Adelaide M. Lusambili[1,2]*, Stefania Wisofschi[3], Terrance J. Wade[3,4], Marleen Temmerman[3,5], Jerim Obure[3]

**1** Institute for Human Development, Aga Khan University, Nairobi, Kenya, **2** School of Business, African International University, Nairobi, Kenya, **3** Centre of Excellence in Women and Child Health, Aga Khan University, Nairobi, Kenya, **4** Department of Health Sciences, Brock University, St. Catharines, Canada, **5** Department of Obstetrics and Gynaecology, Medical College, Aga Khan University, Nairobi, Kenya

* adelaide.lusambili@aku.edu

**Data Availability Statement:** The transcripts cannot be made openly available for data

## Abstract

Research from sub-Saharan Africa indicate that many women experience varied forms of disrespectful maternity care, which amount to a violation of their rights and dignity. Notably, there is little research that sheds light on health care workers (HCWs) training and knowledge of principles of respectful maternity care (RMC). Formulating appropriate interventional strategies to promote the respectful provision of services for women during pregnancy, childbirth, and postpartum period requires an understanding of the current state of knowledge and sources of information on respectful maternity care among HCWs. This paper reports findings from a qualitative study that examined the knowledge and sources of information on the Respectful Maternity Care Charter among HCWs in rural Kisii and Kilifi counties in Kenya. Between January and March 2020, we conducted 24 in-depth interviews among HCWs in rural Kisii and Kilifi health facilities. Data were analyzed using a mixed deductive and inductive thematic analysis guided by Braun's [2006] six stages of analysis. We found that from the seven globally accepted principles of respectful maternity care, at least half of the HCWs were aware of patients right to consented care, confidentiality and privacy, and the right to non-discriminatory care based on specific attributes. Knowledge of the right to no physical and emotional abuse, abandonment of care, and detentions in the facilities was limited to a minority of health care workers but only after prompting. Sources of information on respectful maternity care were largely limited to continuous medical and professional training and clinical mentorship. The existing gap shows the need for training and mentorship of HCWs on the Respectful Maternity Care Charter as part of pre-service medical and nursing curricula and continuing clinical education to bridge this gap. At the policy level, strategies are necessary to support the integration of respectful maternity care into pre-service training curricula.

protection and ethical reasons. Participants are not named in transcripts, but there are sufficient contextual details to allow indirect identification. The consent agreement signed by participants also sets conditions that limit access to authorized people within the study team only. The project will be unable to make the full un-anonymized transcript available, however selected non-identifiable text might be provided to clarify specific points upon request. Further questions about this study should be addressed to: The Aga Khan University Research Office (AKUkenya. researchoffice@aku.edu).

**Funding:** This study was supported by the Government of Canada and Aga Khan Foundation Canada in the form of a grant awarded to MT (7540280). This study was also supported by Aga Khan Foundation Canada's contribution in the development of the study tools. The funders had no additional role in study design, data collection and analysis, decision to publish, or preparation of the manuscript.

**Competing interests:** The authors have declared that no competing interests exist.

**Abbreviations:** AKFC, Aga Khan Foundation Canada; AKU, Aga Khan University; AQCESS, Access to Quality Care through Extending and Strengthening Health Systems; CHV, Community Health Volunteers; DMC, Disrespectful Maternity Care; GoC, Government of Canada; HCW, Health Care workers; IDIs, In-Depth Interviews; LMIC, Low- to Middle-Income Countries; MERL, Monitoring and Evaluation and Research Learning; MNCH, Maternal, Newborn and Child Health; NACOSTI, National Commission for Science, Technology, and Innovation; RMC, Respectful Maternity Care; RMNCH, Reproductive, Maternal, Newborn and Child Health; SSA, sub-Saharan Africa; TBA, Traditional Birth Attendants.

## Introduction

Respectful maternity care (RMC) is a basic human right recognized by the World Health Organization as it has consequences on the health of the mother and child [1–3]. Clearly articulated in the White Ribbon Alliance Charter on RMC [3], the rights of women during childbirth, and the violation of these rights, which constitute disrespectful care, focus on mistreatment and obstetric violence. They include physical and verbal abuse, withholding treatment, lack of consent for treatment, lack of confidentiality and privacy, discrimination, and involuntary confinement [3]. Referred to as disrespectful maternity care in this paper, violation of women's rights during childbirth, while shown to exist globally, is prevalent among sub-Saharan Africa and across low- and middle-income countries (LMICs) [4, 5]. One of the principal concerns of disrespectful maternity care is that it discourages women from seeking facility-based childbirth, placing both the mother and child at an increased risk for poor obstetric outcomes and mortality [6].

To address respectful and disrespectful maternity care, this paper is part of a series of papers from the research conducted by the Centre for Excellence in Women and Child Health (COEWCH) at the Aga Khan University, Kenya, under the [Access to Quality Care through Extending and Strengthening Health Systems (AQCESS)] project which was funded by Global Affairs Canada through the Aga Khan Foundation Canada (AKFC). Our debut paper in this sequel examined the experience of pregnant and postpartum women and community members at health facilities during pregnancy, birth, and postpartum [6]. The research findings reported in that debut paper reported rampant abuse of women by health care workers during pregnancy, childbirth and postpartum care. These findings, are corroborated by other evidence that arise from similar context in sub-Saharan Africa, which have equally alluded that such mistreatment have the negative effect of dissuading women from reaching out for facility based childbirth services [6–10].

Intrigued by these findings, the COEWCH, through the AQCESS project, conducted a follow up study in 2020 that aimed to examine [1] HCWs perspectives of respectful maternity care, [2] knowledge and sources of respectful maternity care, and [3] strategies and measures required to improve respectful maternity care. This present study was conducted to probe the existing gaps in the body of knowledge of RMC as well as to provide insights that could guide interventions to improve the practice of maternity services in rural Kenya. In the preceding publications in this sequel, we reported about HCW's perspectives on factors that contribute to disrespectful maternity care, which included low staffing and overwork of HCWs, social and cultural beliefs within the population, poor health infrastructure, and healthcare worker's attitudes towards indigent women [11].

These previous analyses presented findings on service users' experiences [6] and healthcare influences on RMC practices to help mitigate disrespect and abuse [11]. However, it is also critical to assess the ability of healthcare workers to implement respectful care by first gaining insight into both their awareness of, and understanding of the principles of RMC, and as well as probing about the sources of their information on RMC [3]. Emerging research shows that health care professionals are not aware of the RMC charter principles as shown in a study in Ethiopia [12]. Yet HCWs knowledge and understanding of the principles of this charter is key to their delivery of quality care for childbearing women. While the RMC Charter exists, we found no evidence on strategies and measures in place to ensure its implementation in health facilities in low and middle income settings. Available research seems to suggest that in many settings and through tailored on-the-job trainings, HCWs are trained on some aspects of the charter to improve quality of maternity care [13–15]. Such short trainings improve nurses' and midwives' knowledge and shift staff negative behaviors towards women during labour and

delivery [13, 14]. In Ethiopia, for example, short courses targeted at health care staff on RMC revealed improved knowledge and staff positive behavior on the treatment of pregnant and postpartum women. HCWs reported increased awareness, knowledge, and a shift in their perceptions and attitudes on how they treat women during birth [15]. Similar findings are recorded in Ghana where midwives reported improved communication and relationships with women during labor and delivery immediately after the training [16]. While delivering short courses is key, however, research also seems to suggest that such trainings may only yield temporary results if monitoring of health care professionals after the training is put in place [17]. Structural health system barriers in low and middle income countries such as low staffing, overcrowded facilities, logistics, access to health care services, and lack of equipment and supplies impedes RMC even in settings where staff are well trained [11, 18]. More often, training on RMC is focused on maternity staff and the importance of extending training and capacity to all hospital staff is needed.

Generally, there is limited research on the extent of knowledge and understanding among HCWs about RMC as emerging research in Kenya and sub-Saharan Africa have mainly focused on the perceptions toward respectful care by service providers and service users. While perceptions on RMC is important, a necessary step in formulating strategies to educate and promote adherence to the *Charter* and underlying principles of RMC is to identify current knowledge gaps that exist with respect to RMC and to identify where HCWs access this information. As such, the aim of this paper is to report on health care workers' knowledge and sources of information on RMC among care providers in two rural counties in Kenya. Having knowledge about RMC is only an intermediary to improving user experience of care, improving facility based childbirth as well as reducing intrapartum and post-partum outcomes.

## Methods

### Study context

This qualitative descriptive study was conducted from January to March 2020 in rural Kilifi and Kisii Counties. In this analysis, we purposefully took a descriptive approach in an effort to provide thick descriptions of HCWs narratives on this topic.

Kilifi and Kisii counties were the primary implementation sites of the AQCESS project since 2015. The two counties also bear some of the poorest maternal and newborn health indicators in Kenya. Some of the AKU interventions in these two counties included improving access and use of maternal new child health (MNCH) services by providing gender-responsive service delivery of care at the facilities, including gendered consultation spaces, a gendered workforce, staff training, gendered toilets, refurbishment of old facilities, and the purchase of new hospital equipment. Other project strategies included male engagement in promotion of MNCH and capacity building of Community Health Volunteers (CHVs) to reach out to marginalized populations. Detailed social–cultural characteristics of these contexts including population size, number of hospitals, patient doctor and nurse ratio, and mortality and morbidity levels are detailed in previous publications [6, 11].

### Sampling and recruitment

A total of 24 HCWs were purposively recruited by AQCESS project staff to represent both large, tertiary referral hospitals and small health centers in AQCESS-supported health facilities across Kilifi and Kisii Counties. Only male and female HCWs who had worked for at least one year in these health facilities and had participated in the clinical mentorship programs were included in this research, since they were considered to have firsthand experience about the changes that had taken place in these facilities as a result of the project. To recruit qualified

health care workers as potential study participants, our project staff approached health facility managers in the participating facilities to identify healthcare workers who met the above-described inclusion criteria [18 female and 6 male healthcare workers].

## Interview protocol

All interview guides were in English as all HCWs were fluent in English. See our detailed study guide in our previous publication [11]. Those agreeing to participate were given a courtesy call to confirm their eligibility.

All in-depth interviews (IDIs) with HCWs in Kilifi and Kisii counties were conducted between January and March 2020. Research facilitation was led by a Social Scientist (AL) with the help of research assistants who were unfamiliar with the participants. Research assistants were trained on the study protocol and how to take field notes. The interview guide was piloted with two participants to identify any gaps in content and to assess the length of the interview.

A convenient time for the interview was agreed to in advance and all interviews were conducted after the end of the work shift, during lunch break, or an hour before the start of the shift to avoid interruptions of service delivery during normal working hours. At the beginning of the interview, all HCWs provided written informed consent, a process which entailed explaining the purpose of the study, the risks and benefits associated with participation, consent to audio record, as well as voluntary withdrawals. Interviews were conducted in one of the offices at the health facilities in which the HCWs worked to allow for private and meaningful conversations. Study facilitators used a detailed study guide which was informed by our previous study findings as well as the AQCESS interventions that had been implemented. All interviews were audio recorded and lasted no more than 60 minutes. Participants were asked to voluntarily share their understanding of respectful maternity care as it relates to consented care, respect, right to information, confidentiality, abuse, and privacy. They were also asked about their sources of information on respectful maternity care, and their attitudes and experiences. Where necessary, prompts were used.

## Data management

All audio-recorded interviews were transcribed by a professional company. Transcribed scripts were saved on a secured computer with a password protected drive at the COEWCH Monitoring, Evaluation and Research (MERL) unit. Research staff (AL and SW) listened to the audio recorded interviews while comparing them with the transcribed data to identify any gaps that may have occurred during the transcription process. Once the investigators were satisfied that the transcribed data were complete, all identifying information such as names of individuals and health facilities were removed, and all audio recorded interviews were deleted as per the ethics requirements.

## Data analysis

Data were analyzed using NVIVO 12 software. Employing a blended deductive and inductive thematic analysis, researchers followed Braun's [2006] six stages of data analysis: familiarization, coding, developing categories, themes, and writing [19]. First, AL and SW read all the transcripts to familiarize themselves with the data. Second, an open coding process was conducted on six randomly selected transcripts. The two researchers then compared codes and agreed on a coding framework which they used to code the remaining transcripts. Categories were then developed from the initial codes and merged into themes.

Two data interpretation workshops were held end of March 2021 that brought together the research team, HCWs, facility managers, selected local stakeholders, and AQCESS project

officers from Kisii and Kilifi, and the donor representatives. Through these presentations and reflections on the findings, workshop participants assisted in validating the results and identified gaps and progress in the AQCESS project.

The *Consolidated Criteria for REporting Qualitative studies* (COREQ) checklist [S1 Table in S1 Checklist] was used to report our findings.

## Study ethics approval

Data collection commenced after ethical approval from the Institutional Research Board [IRB] of the Aga Khan University East Africa [reference 2019/IERC-104 (V2)] and the National Commission for Science Technology and Innovation (NACOSTI/P/19/2768) on 03/December 2019. The research team also obtained permission for the study from the local governments and the local community leaders, health facility managers, and all study participants.

## Findings

Our analysis of data indicates that HCWs working in rural Kilifi and Kisii health facilities did not have adequate knowledge or training on all seven globally accepted principles of respectful maternity care. HCWs reported having knowledge about only two or three categories of respectful care. Reported sources of information for RMC included formal and continuous professional education and exposure through clinical experiences. Section 1 of the findings presents both the HCWs' knowledge and understanding of RMC and Section 2 identifies HCWs' sources of information. The data presented in each section and sub-section contain quotes from both male and female HCWs but were not identified by gender to ensure confidentiality.

### Section 1: HCWs knowledge on respectful maternity care

We used Bowser and Hill [2010] seven rights categories of disrespect and abuse to analyze participant's reflective narratives of their knowledge [20]. Table 1 illustrates the process of analysis

**Table 1. Alignment of study findings with categories of disrespectful care and corresponding respectful maternity care charter rights [adapted from the respectful maternity care charter][1,2].**

| | Category of Disrespect and Abuse | Corresponding Respectful Maternity Care Charter Right | Number [percentage] of participants referencing corresponding RMC right N = 24 |
|---|---|---|---|
| 1 | Non-consented care | Right to information, informed consent and refusal, and respect for choices and preferences, including companionship during maternity care | 18 [75%] HCWs promptly mentioned this right without prompting. |
| 2 | Non-confidential care | Right to confidentiality and privacy | 12 [50%] HCWs mentioned this right without prompting. |
| 3 | Discrimination based on specific attributes | Right to equality, freedom from discrimination, equitable care | 12 [50%] HCWs mentioned this right without prompting. |
| 4 | Physical abuse | Right to freedom from harm and ill treatment | 7 [29%] HCWs mentioned this when prompted. |
| 5 | Non dignified care [including verbal abuse] | Right to dignity and respect | 7 [29%] HCWs mentioned this when prompted. |
| 6 | Abandonment or denial of care | Right to timely healthcare and to the highest attainable level of health | 5 [21%] HCWs mentioned this when prompted. |
| 7 | Detention in facilities | Right to liberty, autonomy, self-determination, and freedom from coercion | None [0%] of the HCWs mentioned this right and even when prompted they did not seem to have heard about it. |

[1] Respectful Maternity Care: The Universal Rights of Childbearing women, Table 1: Tackling Disrespect and Abuse: Seven Rights of Childbearing Women. [https://www.who.int/woman_child_accountability/ierg/reports/2012_01S_Respectful_Maternity_Care_Charter_The_Universal_Rights_of_Childbearing_Women.pdf. Accessed July 26, 2021].

[2] Note: The ordering of rights in this table is based on the percentage of reported familiarity by Health Care Workers [HWCs] identified during interviews.

based on the White Ribbon Alliance guidance and adapted from the World Health Organisaton (WHO) *Respectful Maternity Care: The Universal Rights of Childbearing Women report* to indicate the percentage of HCW participants familiar and more conversant with in their practice for each of the RMC rights [1].

The majority of the HCWs were quick to reflect on issues around consented care, confidentiality, and discrimination. However, the principles of *abandonment of care* and *detention in facilities* were rarely understood by many of them and were only mentioned if the interviewer probed the HCWs. A few HCWs noted that they had heard of physical abuse, although this did not come up unprompted from their narratives. However, when specifically asked about it during the interview, many HCWs indicated it was a violation of women's basic rights and that they had heard of it. As such, it may be perceived more generally as a basic human right and participants did not feel it needed to be identified as a specific right to RMC.

**1. Good understanding on the right to non-consented care, non-confidential care and freedom from discrimination.**   *[a] The right to informed consent for care.* Findings allude to the importance of informed and consented care among HCWs knowledge and understanding. The majority of HCWs interviewed highlighted the importance of obtaining consent from patients either orally or in writing. HCWs also highlighted the significance of providing information to empower and enable women to exercise their rights in informed decision-making in relation to their care. The HCWs also reported that, increasingly, patients have more information and therefore take the responsibility to ask questions and share their health concerns with HCWs.

*. . .there is no procedure we can do to this mother without the mother alone consenting. There are those ones which they consent, there are those ones they can give orally, . . . there are those ones [mothers] which they can give by signing, especially like if we are taking the mother to theatre, if for example we say that there's a complication prone to labour and such things, we need to tell either the guardian or the mother to consent, or the husband. Mostly we involve the husband if we are taking the mother for the exam for CS [C-section], she has to consent by signing. Yes. . .*

IDI, Nurse, Kisii

*Any procedure that we are doing we make sure we get consent from the mothers. . ..*

IDI, Nurse, Kilifi

Both Kilifi and Kisii counties are rural regions with low literacy levels. As such, the revelations from HCWs provided in these interviews about the importance of consented care is an indication of the value that medical and continuous education training may put on this aspect of training.

*[b] The right to privacy and confidentiality.* About half of the HCWs who participated in the research identified confidentiality and privacy as an important aspect of RMC. Confidentiality was mentioned with respect to not disclosing client's personal information while privacy was highlighted as being important in the provision of care during labour and after birth as illustrated below.

*. . .the issue of confidentiality is one of the pillars in our health services. In fact, confidentiality is the key. We have even taken an oath. We don't disclose the patients; I mean the clients information. . .*

IDI: Nurse–Kisii

*. . .Respectful Maternity Care is care given to mothers who have come to deliver, which is, care given to mothers who have come, . . . to deliver and their neonates during first stage, during birth, and after birth. Which [is to] maintain their confidentiality, their privacy and the care which they need and for them to be, to feel comfortable, and to appreciate the services which are being given to them. . .*

IDI: Nurse–Kisii

As the above quotes from HCWs in Kisii illustrate, confidentiality and privacy were often seen as synonymous with 'quality care'. HCWs saw it as their duty to maintain confidentiality of the mothers, having taken an oath not to disclose any information concerning their care. HCWs also observed that confidentiality was not only important during birth but also in the postnatal period.

In Kilifi, HCWs reported similar views that confidentiality and privacy were synonymous to quality of care.

*Respectful Maternity Care, this is care that is given to the mothers or women who have come to the maternity in labour, and we give the care that is quality. . . taking care of their privacy. . ., when you are taking care of them you ensure there is privacy.*

IDI: Nurse–Kilifi

In the rural Kenyan context, historically, some women have shunned away from using facility-based care for birth due to the perception that they are shouted at and humiliated by female health professionals and that their data is not kept confidential by the HCWs. The revelations by HCWs may indicate a shift toward practice that focused on greater privacy and confidentiality of patients. Although privacy and confidentiality were described as the pillar of their care, HCWs did not mention some of the tangible ways in which they work to preserve privacy of the mothers.

*[C] The right to freedom from discrimination.* About half of the HCWs interviewed also highlighted the importance of providing maternity care that is free of discrimination such as on the basis of culture or age. Communication with the mothers and family members to improve their birth preparedness in a respectful, encouraging and re-assuring manner was underscored by the HCWs in this study, as shown in the findings below from Kilifi.

*. . .Respecting cultures is, okay, an obligation, I can say it is a right, it is okay,. . .depending on the culture itself because some of them [service users] they are okay, they have scientific ideological rationale and justification that they can be adhered to. So, if in a positive way, yes, if whatever she believes in culturally is something that has also an articulation in medical value, then we deal with it positively. . . we respect their cultural values but at the end of it depending on how it is then we have to interact. . .*

IDI, Nurse, Kilifi

In Kenya, there are many cultural practices that rural women may adhere to during pregnancy and birth such as the use of herbs, or the necessity of a female birth attendant for delivery. These cultures are informed by traditional and religious beliefs and, often, some women fail to use health facility services for birthing due to poor health care attitudes towards these beliefs while providing maternity care. HCWs in rural Kilifi where there is a multiplicity of cultural practices showed their willingness to respect and embed positive cultural beliefs of mothers during pregnancy. In addition, HCWs also reported the importance of involving and sharing information with relatives in the care of mothers during birth.

*It is quality care and it involves not only the mother but also the relatives, in case of any procedure we involve all parties. . . for the relatives, they have the right to know what you are doing to the mother, so in case of any procedure or any concern you involve them. . . so in case of anything they are to watch, or but there is privacy not really exposing the mother. . . she will be involved in any activity that will be done.*

IDI, Nurse, Kilifi.

**2. Limited knowledge on the right to dignified care, physical abuse and abandonment of care.** *[a] The right to dignity and respect.* Overall, HCWs reported a low awareness of the principle of dignity and respect of patients. Findings, largely among HCWs from Kilifi, indicate that some HCWs had some knowledge of the principle of dignity and respect in providing maternity care. This was mentioned with respect to the behavior of HCWs treating mothers with care and professionalism. Commonly used terms under this theme included: 'being attended to in the right way', 'being polite' and 'showing respect'.

*. . .Respectful Maternity Care is where when a mother comes to a facility maybe she needs a certain service, she is attended to in the right way. In the right way means that she is welcomed, she is provided with the services she required, then she comes out of that facility when she feels like she is comfortable and she has been accorded all the services that she required, . . .*

IDI: Nurse–Kilifi

*. . . we should also respect her, her views that care. . . communication, you should be polite, not abusive, and you should communicate in a way that the client understands"*

IDI: Nurse–Kilifi

*"Okay, this is a quality care given to pregnant mothers during their entire period of their pregnancy. . .it entails giving care during the antenatal period, during the birth and also after the birth."*

IDI: Nurse–Kilifi

*[b] physical abuse and abandonment of care.* The majority of the healthcare workers did not mention patients' rights with respect to abuse until they were prompted by the interviewer, nor did they appear to be generally aware of patients' rights with respect to either abandonment of care or detention in facilities. In fact, only seven of the 24 HCWs (29%) referenced abuse even when prompted about it. They noted that this is a key priority in providing respectful maternity care to mothers–care that is free of verbal, psychological, and physical abuse. Moreover, none of the HCW mentioned the right of patients against detention in the facility while only five (21%) referenced the right of mothers against the denial and abandonment of care. They acknowledged that mothers seeking maternity care are in a position of vulnerability. By placing their trust in the hands of the HCWs, they deserved to be provided with the best possible care without risk of abuse.

*. . .My thought about the abuse of women verbally or psychologically or physically or whatever manner is, . . . is unprofessional and unethical. It is even a sin and a crime in the eyes of God if I may put it. Constitutionally it is a crime.*

IDI: Nurse–Kilifi

*. . .by meaning respectful it means we do not abuse the mothers either physically, or emotionally, so it is quality care*

IDI: Nurse–Kilifi

In the health facilities in the two Counties, there had been numerous reports of physical abuse and abandonment of care by patients. In our previous research on disrespectful care, we identified several patients and family members who reported that women were being physically and verbally abused and where health care workers abandoned care [6]. HCWs' low awareness of Respectful Maternity Care Charter right on abandonment of care may suggest the normalization of this practice and, as such, they may not see the necessity of upholding this practice.

## Section 2: HCWs' sources of information on respectful maternity care

HCWs were asked to reflect and share their views on whether they had learned about respectful maternity care and where they received this information from. Our analysis shows that, while some HCWs learned this information during their clinical education as part of their program curriculum, the majority of HCWs reported learning about respectful maternity care during their nursing work experience and through continuous mentoring medical training organized at the facilities in which they worked–either by development agencies and programs such as AQCESS or by the Ministry of Health. Table 2 summarizes the reported sources of information and frequency of reporting as informed by the HCW interviewees.

**1. Formal education and continuous professional trainings through mentorship.**

1. Formal education during medical training and continuous professional development courses through local agencies instilled values among health care workers on the need to offer respectful maternity care to patients. HCWs who had worked in facilities where programmatic work such as AQCESS was undertaken to upgrade healthcare worker's skills and knowledge through a mentorship study reported the valuable contribution of such programs towards their providing respectful maternity care.

**Table 2. Summary of identified sources of information on Respectful Maternity Care (RMC) by health care workers.**

| Sources of information on RMC | Number [percentage] of participants referencing source N = 24 |
|---|---|
| 1. Through varied sources of **on-the-job training [post-education qualifications]**.<br>• **Yearly continuous professional training**<br>• **Through AQCESS and other projects where trainings were provided on the provision of quality health services**<br>• **Through training from the Ministry of Health** | 13 [54%] |
| 2. Through work experience gained through many years of working at health facilities **interacting with service users and learning/ understanding the importance of offering quality care to pregnant women as well as offering culturally appropriate services to different women across different age and religious groups. For example, the need for a female doctor to attend to a Muslim woman during birth and/or being empathic and non-discriminatory to younger, older, and other marginalized [e.g., disabled] women**. | 8 [33%] |
| 3. Nursing and Medical education | 2 [8%] |

*. . . Yes, and then at the facility level especially for us people who have been in the AQCESS project area, we have had a program what we call mentorship, . . .under AQCESS, we have had trainings, we have had, mentorship, yes, on job trainings, yes, these have greatly instilled value on how we care and to tackle the clients in respectful ways. . .*

IDI, Nurse, Kisii

Our data further show that, although learning about respectful maternity care may not generally be taught as part of the medical curriculum in Kenya, HCWs' may learn aspects of these principles at different times on their professional journey—from college and other professional trainings.

*. . .Okay, I learned from school when I was in college then I've worked in a private facility before I was employed to the county government, so I learned some few things from there. Yes, from college experience and wherever I was working. . .*

IDI, Nurse, Kisii

*. . .in the nursing school basic training, I got this information I had also upgraded, so I think as a nurse, as a midwife this thing are being repeated once again. I mean, it's a repeat of the same we learn them, and we continue to relearn them again. . .*

IDI, Nurse, Kilifi.

**2. Clinical work experience.**

1. Across the two research sites, both work and personal experiences were highlighted as opportunities for continuous learning and personal development related to principles of respectful maternity care. In fact, one health care provider described how her personal experience with maternity care influenced her understanding of respectful maternity care.

*. . .Ok from experience, I have worked for many years mostly on case of maternity . . . I have worked at the sub county here, mothers came up and talk even themselves, this nurse is good. This nurse is treating us well, attends to you well. Any question you ask is there to answer you. . .*

IDI, Nurse, Kisii

*. . . the information I come to know about it, first because if you take yourself as the mother who has come to deliver, and then you get handled abusively, you ideally will not come back to that hospital, you will not take your relative there. So, the information from the school, and then when we went for attachment, you know there were those old mothers, who used to be shouted at and beaten. . . we didn't copy that behavior, okay, or I didn't, because if am handled that way, I will not go for that service, so from my own experience, I know you must handle that mother, as yourself would like to be handled. . .*

IDI, Nurse, Kisii

## Discussion

While there is a growing body of research on the perspectives and perceptions of service providers toward RMC, there is little understanding of the level of knowledge and understanding

of care among practitioners as well as the sources of this knowledge. The current study addresses this important gap in existing research by examining two rural counties in Kenya on HCWs' knowledge and understanding of RMC. Overall, HCWs identified a number of salient issues regarding the importance of delivering respectful maternity care. They also indicated that much of their knowledge and training in RMC comes after their formal health care education, gained mainly from post training work experience and their participation in continuous education training and other capacity-building programs. These findings are consistent with findings from studies conducted in Sudan [13]; Ghana [14], and Ethiopia [15]. Under the Kenyan devolved government, gaps exist in teaching RMC principles to health care professionals after medical college. The responsibility lies with the various health facilities to ensure that their staff are trained. Even where staff are provided with short courses, they do not cover all the principles as envisioned in the charter. This raises questions as to how the WHO can ensure that policies implemented at the macro level can be reinforced on the ground and benefit the most marginalised populations including timely trainings for HCWs working with these populations.

With respect to RMC, half or more of the HCWs demonstrated good knowledge and understanding of some of the principles that are necessary to provide high quality, respectful care, including consent for care, confidentiality, and non-discrimination. However, with respect to the White Ribbon Alliance Charter on RMC, there were some notable knowledge deficits that may have hamstrung good practices around the RMC. Few HCWs among those interviewed identified patients' rights with respect to freedom from physical abuse, undignified care [e.g., verbal abuse], and abandonment of care. And even among those few who identified these rights of the patient only did so after prompting by the interviewer. Consistent with a study conducted by Jolly et al. [12], this finding is not surprising given that our previous research on RMC that focused on pregnant and postpartum women and community members indicate that physical and verbal abuse as well as neglectful care is prevalent in the two counties where this study was conducted [6–9]. This finding may indicate the need for more training and information sharing on issues surrounding physical, emotional, and verbal abuse. As well, the right to liberty, autonomy, self-determination, and the freedom from arbitrary detention was not mentioned at all by any of the HCWs interviewed. This could be due to the fact that there is a Free Maternity Policy in Kenya [21] that prohibits the detention of mothers and their babies in public hospitals for non-payment of fees. Furthermore, the right of the newborn to be with their parents or guardians was not identified in this study. This omission is likely a result of the study focus and narratives based more on the mothers with little reference to newborns. As the well-being of a woman and her newborn are interconnected with the aim to address any unnecessary marginalization during childbirth and thereafter, subsequent research should expand its focus beyond the mother to the mother-child dyad [1].

An additional important finding from our study is that the overwhelming majority of HCWs reported learning about RMC through work experience and continued medical and nursing education, while only a few reported learning about it during their formal programs of study. This underscores the urgent need to integrate RMC training into the National Medical and Nursing Curricula to emphasize its importance in providing maternal care. Moreover, it identifies an opportunity to use continuing medical and nursing education, mentorship, and supportive supervision for HCWs with targeted topics emphasizing RMC to address deficits among current HCWs and to complement and reinforce their formal training. This will ensure a greater awareness among currently practicing HCWs about the importance of this aspect of care. WHO impresses upon health care providers to put interventions in place that incorporates clients' aspirations and preferences including cultural aspects of their communities [22]. Respectful maternity care is everyone's responsibility as abuse of childbearing mothers

happens at different levels. These findings raise questions not only for the HCWs but also for the general public, mothers, and all allied hospital staff to be educated in respectful care.

## Policy implications

The findings of this study have several implications for policy and practice, providing a useful roadmap when developing strategies to implement a rights-based approach to maternal and newborn care. Health facilities should implement clear and measurable strategies regarding RMCs in their facilities, as informed by the Charter. This could include the development of standard operating procedures, education, and the development of communication materials on RMC for both HCWs and women made available at all health care sites. This study also identifies a need to co-develop strategies with the Ministry of Health and support integration of RMC through both pre-service education and in-service training of HCWs. There is also a need to promote facility based reflective practice and routine reviews of care during labour and childbirth to identify areas for improvement and enhanced quality. The fact that these findings are from the AQCESS project intervention sites makes these results all the more important because one expects that these sites would be more up-to-date on respectful and disrespectful care through their participation with the program. If these sites also have deficits in RMC knowledge, it suggests a larger, systemic issue that is likely best addressed through national initiatives.

Targeted interventions on RMC should also aim to raise awareness among HCWs on the Charter rights and principles that were not mentioned by participants in this study. This is important as these Charter rights are indivisible, human rights and should be viewed from a holistic perspective. These include being provided with equitable and non-discriminatory care and the best possible level of health care for mothers and newborns as well as being treated with respect and being involved in decisions related to their care.

## Future research

We recommend further qualitative and quantitative research to probe all components of RMC as defined by the WHO to help better understand, identify, and prioritise aspects of RMC that require intervention at the service delivery levels. Research could help us to identify barriers and opportunities for interventions. In addition, a national survey of health care workers [HCWs] that both identifies the familiarity with the Charter of Rights of childbearing women and ranks the importance and readiness of sources of information would provide policy makers with insights on the availability and use of information on RMC. Mixed methods research that focuses on both rural and urban settings, including informal settlements, could provide views of HCWs in varied contexts, thus, allowing policy makers to develop tailored training tools to promote RMC among different groups of health care workers.

## Study strength and limitations

As shown from the findings above, HCWs were able to identify only some of the tenets of RMC as postulated by the WHO [2018] without prompts. This may indicate a methodological flaw in our design, which needs to be factored into future research to allow HCWs to reflect on all the components of RMCs in their practice. The study is also limited to rural settings, and in counties where there was an ongoing intervention that included staff mentorship at the facility levels. Thus, it is difficult to generalize these findings to other geographically disconnected rural settings where there are no interventions, to populations that still use traditional birth attendants (TBAs), and to HCWs in urban settings.

As a descriptive, exploratory study that examined knowledge of RMC among HCWs in Kenya, this study begins to close major research gaps in Kenya and sub-Saharan Africa (SSA) where topics relating to quality of care and training of health workers are poorly studied. The findings can inform future studies across a variety of different geographical settings, development of training manuals and guidelines in the maternal health facilities, as well as inform policies to guide the respectful care of pregnant and postpartum women in rural health facilities.

## Conclusions

Our findings indicate that while HCWs in Kilifi and Kisii counties of Kenya possess good knowledge of some of the principles of RMC, there remain significant knowledge gaps. This could be addressed by promoting a rights-based approach to maternal and newborn health through the inclusion of RMC in pre-service curricula and clinical training of HCWs in line with global standards and best practices. As the majority of HCWs report only learning about RMC through work experience and continued medical and nursing education, integrating RMC into formal, pre-service training curricula would ensure a greater awareness of, and attention to, the rights of pregnant and postpartum women resulting in greater use of facility-based maternity care and improved health and mortality among mothers and babies.

## Supporting information

**S1 Checklist. Consolidated criteria for reporting qualitative studies [COREQ] checklist.** (DOCX)

## Acknowledgments

We acknowledge the Aga Khan University Monitoring and Evaluation Unit (MERL) unit for coordination and execution of this research. We acknowledge field project managers and research assistants in both Kisii and Kilifi research sites for logistical support during the field and data collection respectively. We are grateful to the AQCESS project endline consultant. Many thanks to our local partners in Kilifi and Kisii and to all the HCWs who participated in this study. We acknowledge the enormous assistance offered by the Aga Khan Foundation Canada (AKFC) team in the development of the study tools. Thanks to Carol Jaka for comprehensive review of our first draft. Lastly, thanks for the funding support from the Government of Canada and Aga Khan Foundation Canada.

## Author Contributions

**Conceptualization:** Adelaide M. Lusambili, Marleen Temmerman.

**Data curation:** Adelaide M. Lusambili, Stefania Wisofschi.

**Formal analysis:** Adelaide M. Lusambili, Stefania Wisofschi, Terrance J. Wade.

**Funding acquisition:** Marleen Temmerman.

**Investigation:** Adelaide M. Lusambili.

**Methodology:** Adelaide M. Lusambili, Stefania Wisofschi.

**Project administration:** Marleen Temmerman.

**Supervision:** Adelaide M. Lusambili, Stefania Wisofschi, Marleen Temmerman, Jerim Obure.

**Validation:** Adelaide M. Lusambili, Stefania Wisofschi, Terrance J. Wade, Marleen Temmerman, Jerim Obure.

**Visualization:** Adelaide M. Lusambili, Terrance J. Wade, Jerim Obure.

**Writing – original draft:** Adelaide M. Lusambili, Stefania Wisofschi, Terrance J. Wade.

**Writing – review & editing:** Adelaide M. Lusambili, Stefania Wisofschi, Terrance J. Wade, Jerim Obure.

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
