## [Decision Letter · Decision Letter 0]

13 Oct 2021

PGPH-D-21-00476

A Qualitative Enquiry of Nurses Voices on Respectful Maternity Care (RMC) Principles

Dear Dr. Lusambili,

Thank you for submitting your manuscript to PLOS Global Public Health. After careful consideration, we feel that it has merit but does not fully meet PLOS Global Public Health’s publication criteria as it currently stands. Therefore, we invite you to submit a revised version of the manuscript that addresses the points raised during the review process.

Along with the reviewers, I commend you for an article that takes on a critical topic. In revising your manuscript please pay careful attention to Reviewers #1 and #2's comments about the additional detail needed in the methods section. As the reviewers suggest, following the COREQ guidelines for reporting qualitative research and providing the interview agenda/coding scheme as appendices would significantly strengthen the paper and improve the likelihood of replication in other settings.

We look forward to receiving your revised manuscript.

Kind regards,

Nicola Hawley

Academic Editor

Journal Requirements:

1. Please include a copy of the interview guide used in the study, in both the original language and English, as Supporting Information, or include a citation if it has been published previously.

2. Please provide additional details regarding participant consent. In the ethics statement in the Methods and online submission information, please ensure that you have specified what type you obtained (for instance, written or verbal, and if verbal, how it was documented and witnessed).

3. During your revisions, please note that a simple title correction is required: an apostrophe should be included after Nurses (i.e. ...of Nurses' Voices...). Please ensure this is updated in the manuscript file and the online submission information.

4. We ask that a manuscript source file is provided at Revision. Please upload your manuscript file as a .doc, .docx, .rtf or .tex. If you are providing a .tex file, please upload it under the item type ‘LaTeX Source File’ and leave your .pdf version as the item type ‘Manuscript’.

5. We notice that your Appendix are included in the manuscript file.  Please remove them and upload them  with the file type 'Supporting Information'  . Please ensure that all Supporting Information files are included correctly and that each one has a legend listed in the manuscript after the references list. 

6. Since your data is not available for proprietary reasons, please explain via email why the data is not available. Please also include the contact information for the third party organization that should be contacted should other researchers want to request access to this data and please include the full citation of where the data can be found. We also request that you verify with us via email that any researcher will be able to obtain the data set in the same manner that the you have obtained it. If you feel you are unwilling or unable to adhere to this policy, please explain your reasons by return email and your exemption request will be escalated to the editor for approval. Your exemption request will be handled independently and will not hold up the peer review process, but will need to be resolved should your manuscript be accepted for publication. One of the Editorial team will be in touch if they require more information.

7. Please amend your] detailed Financial Disclosure statement. This is published with the article, therefore should be completed in full sentences and contain the exact wording you wish to be published.

i). State the initials, alongside each funding source, of each author to receive each grant.

ii). State what role the funders took in the study. If the funders had no role in your study, please state: “The funders had no role in study design, data collection and analysis, decision to publish, or preparation of the manuscript.”

Additional Editor Comments (if provided):

Reviewers' comments:

Reviewer's Responses to Questions

**Comments to the Author**

1. Does this manuscript meet PLOS Global Public Health’s publication criteria? Is the manuscript technically sound, and do the data support the conclusions? The manuscript must describe methodologically and ethically rigorous research with conclusions that are appropriately drawn based on the data presented.

Reviewer #1: Yes

Reviewer #2: Partly

Reviewer #3: Yes

2. Has the statistical analysis been performed appropriately and rigorously?

Reviewer #1: Yes

Reviewer #2: N/A

Reviewer #3: N/A

3. Have the authors made all data underlying the findings in their manuscript fully available (please refer to the Data Availability Statement at the start of the manuscript PDF file)?

Reviewer #1: Yes

Reviewer #2: Yes

Reviewer #3: Yes

4. Is the manuscript presented in an intelligible fashion and written in standard English?

Reviewer #1: Yes

Reviewer #2: Yes

Reviewer #3: Yes

5. Review Comments to the Author

Reviewer #1: PLOS Global PH Review: A Qualitative Enquiry of Nurses Voices on Respectful Maternity Care (RMC) Principles

An interesting study that provides important findings regarding respectful maternity care. Some comments to clarify and strengthen the manuscript below.

Backgrounds

Pg 2.

- Consistency re use of antenatal. In the manuscript, there is ‘ante-natal’, ‘ante natal’ and ‘antenatal’ – use what is used in the journal usually

- Use HCWs in brackets first then use acronym

Methods:

- More information about the study setting in the study context – are these counties urban, rural, remote, population size, number of births a year

- How many health facilities were staff recruited from, where these facilities large tertiary referral hospitals or smaller rural hospitals/health centres?

- Note that the findings are reported according to the consolidated criteria for reporting qualitative research (COREQ) checklist – as you then provide as an appendix

Data collection:

- Can you provide a copy of the interview guide as an appendix?

- Were HCWs quizzed on RMC components or did they offer this information voluntarily – what questions were they ask? – another reason to provide a copy of the interview guide

Data management:

- Did participants have an opportunity to review transcripts?

Analysis:

- Can you provide a copy pf your coding framework in the appendix?

- Given many of your results are presented quantitatively, it sounds very much like a content analysis? Might be worth describing how this quantitative analysis took place

Findings:

- Findings are reported against difference categories of quality care – would be useful to describe these in the background and in the methods discuss how these categories were applied to the findings of the qualitative analysis.

- On page 6 – there is some more description about the rural regions with low literacy – would be good to embed this earlier on in the study setting/context

- Also page 6, line 193 should be were poorly understand

Discussion:

- In the discussion, it would be good to have greater unpacking of enablers and barriers to RMC

- Study strength and limitations subheading should be bolded

References:

- Consistency with respect to WHO or World Health Organization

- Review some of the references for grammar etc, inconsistent spacing between references

- Reference 13 is highlighted

Reviewer #2: The manuscript represents an important and timely topic of international interest, and is generally well presented. However, it has a number of weaknesses.

1. The authors reveal that this paper stems from a larger programme of research, in which there has been a series of papers. I do wonder whether the findings presented in this manuscript would have been better combined with a previous paper to provide greater evidence and critical discussion of the key points. By dissecting the work, these findings have less impact, as without referring back to the previous papers, it is difficult to see the whole picture.

2. The introduction could be strengthened by inclusion of recently published papers, with a clearer justification for the current study.

3. The manuscript does not follow COREQ reporting standards, thus there are many omissions. I would advise that this is followed. The methodology is limited; for example, there is no information on why a descriptive approach was adopted, why the sample size of 24 was chosen [was data saturation sought and gained?], what informed the interview guide? was the guide piloted?, and what was the recruitment strategy and process?

4. The study is based on the 2012 RMC Charter, but there is no rationale for using this particular charter as opposed to more recent ones, that have evolved over the last decade, e.g. WRA Charter 2018. This needs discussion.

5. On page 3, line 134, the authors mention a 'team of researchers'. How many researchers? What were there backgrounds [reflexivity is important]? How were they trained?

6. Where the interviewers known to the interviewees? Given that the interviews took place in the interviewees work place, and the subject area is very sensitive, this is particularly important. What, if anything, was done to mitigate against provision of desirable answers and/or any consequences of participation?

7. Data analysis. This section does not include the use of NVIVO, whereas the abstract does. This needs correcting.

8. Findings. In table 1 and 2, it is not usual to include percentages with small numbers.

9. Findings. No characteristics have been provided for the included participants, therefore it is impossible to determine how transferable they are. A table would be needed to include participant factors such as years of professional training, professional group, prior RMC training, are of residence [Kilifi or Kisii] etc.

10. Findings. Although this is a descriptive qualitative study, one would still expect some further discussion around the quotes. In some places quotes are listed.

11. The discussion could be strengthened by better integrating the findings into the current evidence base, and more explicitly highlighting what this paper brings that is new.

12. The future research section lacks focus. Based on the findings, in combination with what we know from existing literature, we need research that not only increases our knowledge but translates that knowledge into action. This section needs revisiting.

13. Limitations. The sites are atypical as they were programme implementation sites. This needs bringing out more in the limitations

14. In places the language is not woman-friendly, such as use of the word 'patient' instead of 'woman', 'deliver' instead of 'birth' and 'antenatal and postnatal women', instead of 'women in the antenatal or postnatal period.'

Reviewer #3: Thanks for the opportunity to review this manuscript on “Nurses Voices on Respectful Maternity Care (RMC) Principles.” This is an excellently written manuscript. After reading this manuscript, I have a few suggestions to make to the authors. The goal of the suggestion is to help them add an important aspect to their qualitative method.

It will be good when the authors add a section on how they dealt with the issue of trustworthiness (the role of the researchers and the issue of reflexivity) in their data collection and management. What’s the professional/academic background of the authors? Could their background have any influence on the way the data was generated and managed?

Thank you.

6. PLOS authors have the option to publish the peer review history of their article (what does this mean?). If published, this will include your full peer review and any attached files.

**Do you want your identity to be public for this peer review?** For information about this choice, including consent withdrawal, please see our Privacy Policy.

Reviewer #1: No

Reviewer #2: No

Reviewer #3: No

---

## [Decision Letter · Decision Letter 1]

11 Apr 2022

PGPH-D-21-00476R1

A qualitative Inquiry of Health Care Workers' Narratives on Knowledge and Sources of Respectful Maternity Care (RMC) Principles.

Dear Dr. Lusambili,

Thank you for submitting your manuscript to PLOS Global Public Health. After careful consideration, we feel that it has merit but does not fully meet PLOS Global Public Health’s publication criteria as it currently stands. Therefore, we invite you to submit a revised version of the manuscript that addresses the points raised during the review process.

Thank you for your attention to the comments of the prior reviewers. They have just a few additional suggestions, which I agree will strengthen the submission. 

We look forward to receiving your revised manuscript.

Kind regards,

Nicola L. Hawley

Academic Editor

Journal Requirements:

1. Your co-authors:

Stefania Wisfoschi -swisofschi@gmail.com

Terrance J Wade -twade@brocku.ca

Jerim Obure -Jobure@ijm.org

,have not confirmed authorship of the manuscript. We have resent them the authorship confirmation email; however please check that the above email address for them is correct and follow up personally to ensure they confirm. 

Please note that we cannot proceed your manuscript  until we have received confirmations from all co-authors.

Additional Editor Comments (if provided):

Reviewers' comments:

Reviewer's Responses to Questions

**Comments to the Author**

1. If the authors have adequately addressed your comments raised in a previous round of review and you feel that this manuscript is now acceptable for publication, you may indicate that here to bypass the “Comments to the Author” section, enter your conflict of interest statement in the “Confidential to Editor” section, and submit your "Accept" recommendation.

Reviewer #2: (No Response)

2. Does this manuscript meet PLOS Global Public Health’s publication criteria? Is the manuscript technically sound, and do the data support the conclusions? The manuscript must describe methodologically and ethically rigorous research with conclusions that are appropriately drawn based on the data presented.

Reviewer #2: Yes

3. Has the statistical analysis been performed appropriately and rigorously?

Reviewer #2: N/A

4. Have the authors made all data underlying the findings in their manuscript fully available (please refer to the Data Availability Statement at the start of the manuscript PDF file)?

Reviewer #2: Yes

5. Is the manuscript presented in an intelligible fashion and written in standard English?

Reviewer #2: Yes

6. Review Comments to the Author

Reviewer #2: Thank you for the amendments made to this manuscript.

There are just a few outstanding considerations, which have either not been addressed/considered in the text or response to feedback.

Reviewer 2

Point 3. It was requested that the authors provide rationale for using a descriptive approach, the chosen sample size, and content of interview guide; this has not been done.

The response to whether interviewees were known to facilitators needs to be included in the text, for the readers benefit.

Point 10. Apologies if this point was not clear. The authors state 'We recommend further qualitative research and quantitative research to probe all components of RMC as defined by the WHO to help better understand, identify and prioritise aspects of RMC that require intervention at the service delivery levels'. I was asking for this to be made a little stronger, given that the international community have expressed frustration on the continual exploration of RMC and publishing of rates, whilst interventions have been sparse. Consider whether your findings support future actions.

Point 12. I agree with the response provided and suggest this goes in the manuscript.

7. PLOS authors have the option to publish the peer review history of their article (what does this mean?). If published, this will include your full peer review and any attached files.

**Do you want your identity to be public for this peer review?** For information about this choice, including consent withdrawal, please see our Privacy Policy.

Reviewer #2: No

---

## [Editor Report · Decision Letter 2]

25 Jul 2022

PGPH-D-21-00476R2

A qualitative Inquiry of Health Care Workers' Narratives on Knowledge and Sources of Respectful Maternity Care (RMC) Principles.

Dear Dr. Lusambili,

Thank you for submitting your manuscript to PLOS Global Public Health. After careful consideration, we feel that it has merit but does not fully meet PLOS Global Public Health’s publication criteria as it currently stands. Therefore, we invite you to submit a revised version of the manuscript that addresses the points raised during the review process.

We look forward to receiving your revised manuscript.

Kind regards,

Anteneh Asefa Mekonnen

Academic Editor

Journal Requirements:

Additional Editor Comments (if provided):

Thank you for addressing the comments raised. There are still areas requiring your attention.

Title

"Sources of Respectful Maternity Care (RMC) Principles". This confusing; you could say "sources of information on repectful maternity care recommendations". There are no global principles but a charter which outlines the universal rights of childbearing women

Abstract

"respectful delivery of services" should be "respectful provision of services" and "labour" should be "childbirth". Please also use women-centred languages throughout; for example, use "gave birth" instead of "delivered". For more tips, please read https://www.womenandbirth.org/article/S1871-5192(20)30088-3/fulltext

Instead of mentioning the software used for analysis, describe the method of analysis

Introduction

I see the use of "respectful maternity care" and "disrespect and abuse" interchangeable. However, these two are different in terms of meaning and scope and must therefore be used logically. You could first start with the violation of women's rights during childbirth in health facilities (also termed as disrespect and abuse or mistreatment or obstetric violence) and them move to the strategy to mitigate disrespect and abuse - "respectful maternity care"

There are double spaces here and there.

"In our previous papers, we provided an understanding of service users’ experiences and healthcare influences on these practices." What do you mean here? you mean the contribution of your study? if so, please revise the sentence. Additionally, cite the study

"Yet there is limited research on what HCWs know and understand about RMC" Two points here 1) there is huge difference between knowledge and understanding; be specific to clearly describe what the aim of your study is 2) There is growing body of knowledge on service providers perceptions of respectful maternity care in SSA mainly Ethiopia, Ghana, Kenya, Nigeria, and Tanzania. I guess, your literature review was not thorough.

Methods

Move "Consolidated criteria for reporting qualitative studies (COREQ) checklist (attached) was used to report our findings" to the end of Data analysis sub-section if you are still interested to retain it.

Move "As research on RMC is poorly understood in Kenya, we purposefully took this descriptive approach in an effort to provide thick descriptions of HCWs narratives on this topic." to the Introduction

Move the Ethics Approval to the end of the Methods section

"Study sample participants and recruitment" is redundant and doesn't read well. You could say "sampling and recruitment"

Interview protocol - first start with how the interview guides were developed, contents of the guide, and then go to how these were administered. Maintain the conventional logical order

In the Data analysis, please mention the method of analysis used. From what is written there, I think it is thematic analysis. If so, was it inductive or deductive or both? See also my comment in the Abstract

"Categories were developed from the initial codes and merged into themes as illustrated in table 1 below." - you cannot cite the table here, but in the Results

Results

Please add a descriptive section at the beginning highlighting profile of the key-informants (profession, experience, gender, etc )

How many themes were derived? This should stand out at the beginning.

"The right of freedom from discrimination"should be "the right to......"

Discussion

I found the Discussion This section should include critical reflections on the main findings of the study vis-a-vis the level of mistreatment during childbirth in health facilities in the study area, lessons from respectful maternity care packages implemented in similar settings (see countries mentioned in my earlier comment), other drivers of mistreatment and/or health system challenges in the promotion of respectful maternity care despite adequate knowledge among service provider, etc. Further in depth argument on the mismatch between knowledge and positive attitude is required to substantiate your point.

"antenatal and postnatal mothers" should be "pregnant and postpartum women"

Have a second look at the addition in your Discussion (Line 499 - 503) "The fact that these findings are from the implementation the AQCESS project intervention sites makes these findings all the more important because one would expect that these sites to be more up-to-date on respectful and disrespectful care through participation with the program. If these sites also have deficits in RMC knowledge, it suggests a larger, systemic issue that are likely best addressed through national initiatives." "....all the more important".

Please make a thorough proofreading before we consider this paper for publication.
---

## [Editor Report · Decision Letter 3]

20 Sep 2022

PGPH-D-21-00476R3

A qualitative Inquiry of Health Care Workers' Narratives on Knowledge and Sources of Respectful Maternity Care (RMC) Principles.

Dear Dr. Lusambili,

Thank you for submitting your manuscript to PLOS Global Public Health. After careful consideration, we feel that it has merit but does not fully meet PLOS Global Public Health’s publication criteria as it currently stands. Therefore, we invite you to submit a revised version of the manuscript that addresses the points raised during the review process.

We look forward to receiving your revised manuscript.

Kind regards,

Anteneh Asefa Mekonnen

Academic Editor

Journal Requirements:

1. Please include a copy of the interview guide used in the study, in both the original language and English, as Supporting Information, or include a citation if it has been published previously.

2. Please provide additional details regarding participant consent. In the ethics statement in the Methods and online submission information, please ensure that you have specified what type you obtained (for instance, written or verbal, and if verbal, how it was documented and witnessed.

Additional Editor Comments (if provided):

Thank you for the changes made in response to my comments. The manuscript reads better now. However, The Discussion is still shallow and lacks rigour to substantiate the findings of your study in light with current evidence in the field of respectful maternity care specifically and quality of maternal care generally, of course focusing on providers perspective. I was surprised to see again you have just 13 references, only 3 used in the Discussion. If you are experiencing difficulty identifying relevant resources/articles which could be used to strengthen the Discussion, I can provide you with many. In the first paragraph, be specific - not "care" but "respectful maternity" care.

The Methods sub-section of the Abstract should indicate the method of data analysis and frameworks used (if any)
---

## [Editor Report · Decision Letter 4]

13 Dec 2022

A Qualitative Enquiry of Health Care Workers’ Narratives on Knowledge and Sources of Information on Principles of Respectful Maternity Care (RMC)

PGPH-D-21-00476R4

Dear Dr Lusambili,

We are pleased to inform you that your manuscript 'A Qualitative Enquiry of Health Care Workers’ Narratives on Knowledge and Sources of Information on Principles of Respectful Maternity Care (RMC)' has been provisionally accepted for publication in PLOS Global Public Health.

Best regards,

Anteneh Asefa Mekonnen, Ph.D., MPH

Academic Editor